# Contribution of Protonation to the Dielectric Relaxation Arising from Bacteriopheophytin Reductions in the Photosynthetic Reaction Centers of *Rhodobacter sphaeroides*

**DOI:** 10.3390/biom14111367

**Published:** 2024-10-27

**Authors:** Gábor Sipka, Péter Maróti

**Affiliations:** Institute of Medical Physics, University of Szeged, 6720 Szeged, Hungary

**Keywords:** *Rhodobacter sphaeroides*, reaction center, bacteriophyeophytin, quinone substitution, acidic cluster, thermodynamics, delayed fluorescence

## Abstract

The pH dependence of the free energy level of the flash-induced primary charge pair P^+^I_A_^−^ was determined by a combination of the results from the indirect charge recombination of P^+^Q_A_^−^ and from the delayed fluorescence of the excited dimer (P*) in the reaction center of the photosynthetic bacterium *Rhodobacter sphaeroides*, where the native ubiquinone at the primary quinone binding site Q_A_ was replaced by low-potential anthraquinone (AQ) derivatives. The following observations were made: (1) The free energy state of P^+^I_A_^−^ was pH independent below pH 10 (–370 ± 10 meV relative to that of the excited dimer P*) and showed a remarkable decrease (about 20 meV/pH unit) above pH 10. A part of the dielectric relaxation of the P^+^I_A_^−^ charge pair that is not insignificant (about 120 meV) should come from protonation-related changes. (2) The single exponential decay character of the kinetics proves that the protonated/unprotonated P^+^I_A_^−^ and P^+^Q_A_^−^ states are in equilibria and the rate constants of protonation *k*_on_^H^ +*k*_off_^H^ are much larger than those of the charge back reaction *k*_back_ ~10^3^ s^−1^. (3) Highly similar pH profiles were measured to determine the free energy states of P^+^Q_A_^−^ and P^+^I_A_^−^, indicating that the same acidic cluster at around Q_B_ should respond to both anionic species. This was supported by model calculations based on anticooperative proton distribution in the cluster with key residues of GluL212, AspL213, AspM17, and GluH173, and the effect of the polarization of the aqueous phase on electrostatic interactions. The larger distance of I_A_^−^ from the cluster (25.2 Å) compared to that of Q_A_^−^ (14.5 Å) is compensated by a smaller effective dielectric constant (6.5 ± 0.5 and 10.0 ± 0.5, respectively). (4) The P* → P^+^Q_A_^−^ and I_A_^−^Q_A_ → I_A_Q_A_^−^ electron transfers are enthalpy-driven reactions with the exemption of very large (>60%) or negligible entropic contributions in cases of substitution by 2,3-dimethyl-AQ or 1-chloro-AQ, respectively. The possible structural consequences are discussed.

## 1. Introduction

Photosynthesis utilizes the solar energy absorbed by pigments and transferred in the form of electronic excitation energy to the photosynthetic reaction center (RC) [1,2,3,4,5], common to all photosynthetic species, where it is converted into the charge gradient needed for the long-term storage of solar energy. The RC from the purple bacterium *Rhodobacter* (*Rba.*) *sphaeroides* includes eight pigment molecules, classified into four groups: a special pair of bacteriochlorophylls (P), two accessory BChls (B), two bacteriopheophytins (I), and two ubiquinones (Q) (Figure 1, [6]). While P and B are located at the periplasmic side of the membrane, and Q at the cytoplasmic side, I is placed approximately halfway across the membrane. A slight asymmetry in the structure of the RC leads to a similar asymmetry in the energy levels of the pigments, leading to a highly dominant (A branch) and a non-dominant (B branch) path for charge separation. However, the design can be nearly completely reversed by changes in some crucial amino acids [7]. The proximity between the BChls in the special pair and between the B_A_ and I_A_ pigments suggests a strong interaction between their excited states, leading to the coherent sharing of the excitation energy [8,9]. Similarly, the charge-separated state initially produced (~3 ps) after the decay of P* is a mixture of vibrationally hot P^+^B_A_^−^ and P^+^I_A_^−^ states, which then relaxes within ~1 ps into the cool P^+^I_A_^−^ state [10]. The subsequent electron transfer to Q_A_ with a lifetime of ~200 ps produces a membrane-spanning radical pair P^+^Q_A_^−^ that is stable at a millisecond timescale.

The energetics, kinetics, and pathways of electron transfer centered at I_A_ are exposed to the protein environment and protein dynamics [11]. Several examples of evidence in favor of time-dependent relaxations of the P^+^I_A_^−^ state have been presented (Figure 2). Based on the very fast (1–10 ns) decay of fluorescence, the free energy difference between P* and P^+^I_A_^−^ was estimated to be between 210 and 260 meV, in accordance with the 250–260 meV obtained for the effects of magnetic fields and temperature on the fast (~50 μs) decay of the excited ^3^P triplet state [12]. Quinone replacement studies indicated that RCs may undergo, at the micro- and millisecond timescales, further relaxations, which increase the free energy difference by up to 340 meV after the initial charge separation [13]. However, additional studies are required to test the upper limit of the energetic relaxation of the P^+^I_A_^−^ state and to distinguish this from that of the P^+^Q_A_^−^ states.

The anionic states formed during electron transfer induce not only the reorganization of the protein environment [14], but also the out-of-plane distortion of the chlorin ring [15]. Two distinct conformations of I_A_^•−^ were reported in spectroscopic studies of RC from *Rba.*
*sphaeroides* [16]. Recently, the XFEL (X-ray free electron) structures revealed how the charge separation was stabilized by protein conformational processes [17].

The perturbation of the free energy level of the P^+^I_A_^−^ charge-separated state can be attributed to nearby amino acids, whose contributions have been thoroughly studied [18] but are not fully understood [19]. One highly examined residue is the tyrosine at site M210 because it lies between P and the initial electron acceptors and is a key residue involved in the initial forward electron transfer. Mutant RCs alter the rates of the initial electron transfer (reviewed in [20,21]). Theoretical studies indicate that the magnitude and orientation of the hydroxyl dipole of tyrosine M210 may play an important role in the energetic stabilization of P^+^I_A_^−^ [22]. Indeed, changes in the orientation of this tyrosine’s hydroxyl dipole slowed the electron transfer significantly [23]. Different mutations at M214 near I_A_ decreased the rate of I_A_^−^ → Q_A_ electron transfer, resulting in competition with the charge recombination between P^+^ and I_A_^−^ and in a drop in the overall yield of charge separation [24]. These effects correlated with the volume of the mutant amino acid side chains. Similar results were obtained when the native bacteriopheophytin was replaced by BChl [25]. The perturbations of the electronic environment in the vicinity of P^+^I_A_^−^ and Q_A_ affected both the extent and timescale of the dielectric relaxation. A four-fold decrease in the electron transfer rate from I_A_^−^ to Q_A_ and a similar decrease in the recombination rate P^+^Q_A_^−^ → PQ_A_ were observed in RC lacking the H subunit (LM dimer). This was interpreted as increased flexibility in the region around Q_A_ and as associated shifts in the reorganization energy of the electron transfers relative to that of the native RC [26].

The light-induced P^+^I^−^ and P^+^Q^−^ dipoles are stabilized by the release of H^+^ ions to the periplasmic side, by the uptake of H^+^ ions from the cytoplasmic side of the RC, and by internal proton rearrangements; these processes have energetic as well as temporal and structural constraints. The flash-induced P^+^ state results in very limited proton release [27], and consequently in a very limited energetic stabilization of the dipole. Continuous illumination, however, produces a significant release of protons of up to six protons per RC for 300 s light exposure [28]. The very slow charge recombination and proton release/uptake kinetics that were observed pointed to cascade of lengthy conformational changes, together with the suspected formation of a hydrogen bond network between P^+^ and the periplasmic aqueous phase [29,30]. On the other end of the dipole, the H^+^ uptake in response to the immediate (within 200 ps) formation of P^+^Q_A_^−^ was found to be rate-limited by intraprotein conformational processes [31]. In native RC, the lifetime of P^+^I_A_^−^ (*τ*_I_~13 ns) is much shorter than the time required for protonation to accommodate the dipole. The relatively slow response of protonation prohibits stabilization via proton rearrangement. In AQ-substituted RC, however, the lifetime of P^+^I_A_^−^ is increased significantly to *τ*_I_·(1 + *K*_2_)~1 ms, where *K*_2_ (~exp(Δ*G*_AI_/*k*_B_*T*) denotes the equilibrium constant between P^+^Q_A_^−^ and P^+^I_A_^−^ states [32]. In a small fraction of RCs, the increased lifetime of the P^+^I_A_^−^ state makes it possible to observe the stabilization through protonation.

The kinetics and stoichiometry of flash-induced proton uptake upon the formation of either Q_A_^−^ or Q_B_^−^ are correlated, suggesting that the same residues respond to the generation of both semiquinone species [33,34,35]. The Q_B_ domain is rich in protonatable residues that can account for the uptake of protons in response to the appearance of negative charges on Q_B_ and Q_A_ [36,37,38]. The substoichiometric proton uptake and/or internal redistribution of H^+^ ions arise largely from the same cast of characters in the Q_B_ domain, and this concept is further supported by the lack of ionizable residues around Q_A_ and I_A_ [37,39]. However, the interactions within the ionization states of the individual residues in the complex network and between the acidic cluster and the semiquinones are still not reliably accounted for by the existing computational and spectroscopic methods. Discrepancies between the calculations and experiments (particularly FTIR) remain unresolved for some key residues in the cluster [40,41].

Whether I_A_^−^ can interact with the remote acidic cluster and the possible consequences of this interaction are open questions. There are some experimental indications that the acidic cluster may cooperate with the appearance of the negative charge on I_A_. A Q_A_^−^-induced red shift in the Q_y_ absorption band of I_A_ was observed, which was pH-dependent and showed the involvement of proton transfer in protein relaxation [42]. The absence of the 400 μs component in the relaxation kinetics of the L212Glu-Ala mutant suggested that this residue of the cluster was involved in the relaxation mechanism. The low effective dielectric allows the spread of electric field from I_A_^−^ in the membrane further to the Q_B_ region featured by an unusually high density of ionizable residues with a striking excess of acidic groups. In this way, the cluster of ionizable residues around Q_B_ should contribute substantially to the partial shielding and stabilization of the light-induced P^+^I_A_^−^ dipole. However, direct experimental proof, the mechanism of this, and the extent of the cross-talk remain unclear.

Here, we aim to obtain direct experimental evidence of this interaction. The protonation changes in the acidic cluster around Q_B_ upon the appearance of P^+^Q_A_^−^ and P^+^I_A_^−^ were studied via substitution of the native UQ_10_ with several low-potential analogs of anthraquinone (AQ) and combined measurements of the delayed fluorescence and charge recombination attributed to P^+^Q_A_^−^ → PQ_A_. With AQ, as opposed to UQ, the charge recombination occurs through an indirect pathway via P^+^I_A_^−^ and the rate is sensitive to small perturbations in the free energy states of P^+^Q_A_^−^ and P^+^I_A_^−^. It is revealed that the free energy levels of P^+^I_A_^−^ states were pH-dependent at high (>10) pH values, indicating the participation of residues with high p*K*_a_ values in the interaction between the acidic cluster around Q_B_ and I_A_^−^. Relative to P*, the stabilization of the relaxed (ms) state of the P^+^I_A_^−^ dipole compared to that of the hot (ns) state increased by more than 50% due to this coupling and to the protonation coupled with slow conformational changes.

## 2. Materials and Methods

### 2.1. Bacterial Strain, Media and Chemicals

Both carotenoidless *Rba.*
*sphaeroides* R-26 and the wild-type *Rba.*
*sphaeroides* 2.4.1. strains were inoculated following incubation in the dark for 5–7 hrs in Siström minimal medium. The cells were cultivated anaerobically in 1 L screw-top flasks under a continuous illumination of about 13 W/m^2^, provided by tungsten lamps (40 W), as described earlier in [43]. The assay solution contained 2 μM RCs with 0.03% LDAO, 100 mM NaCl, and a 5 mM buffer, depending on the pH. The following buffers were used: 2-(N-morpholino)-ethanesulfonic acid (MES; Sigma, St. Louis, MO, USA) between pH 5.5 and pH 6.5; 1,3-bis[tris(hydroxymethyl) methylamino] propane (Bis–Tris propane; Sigma) between pH 6.3 and pH 9.5; Tris–HCl (Sigma) between pH 7.5 and pH 9.0; and 3-(cyclohexylamino) propanesulfonic acid (CAPS; Calbiochem) and 2-(cyclohexylamino) ethanesulfonic acid (Ches, Sigma) above pH 9.5.

### 2.2. Anthraquinone (AQ) Substitutions (Figure 2)

To remove the primary ubiquinone (UQ_10_) of native RC from *Rba.*
*sphaeroides*, the standard method [44] was used with some modifications [13,45,46]. The absorption ratio of *A*_280_/*A*_802_ of the Q_A_-depleted RCs was in the range 1.28–1.32. About 90–95% of the RCs were depleted of Q_A_, as confirmed by flash-induced absorption change measurement at 430 nm. In the remaining 5–10% of the RCs, the native UQ_10_ was not removed. The anthraquinone analogs (Fluka) were dissolved in ethanol and used in 10-fold excess. The final ethanol concentration in the solution was <2% (*v*/*v*). Compared to the UQ_10_ reconstituted RCs, the yields of the P^+^Q_A_^−^ formation were 80–90% in AQ (Anthraquinone (AQ), 1-chloroanthraquinone (1-Chloro-AQ), 2-methylanthraquinone (2-Methyl-AQ), 2-ethylanthraquinone (2-Ethyl-AQ), 1-aminoathraquinone (1-Amino-AQ), and 2,3-dimethylanthraquinone (2,3-Dimethyl-AQ)) reconstituted RCs.

### 2.3. Optical Measurements

The photochemical function of the RC and each AQ reconstituted RC was characterized by the near-infrared absorbance spectrum and by the Xe-flash-induced absorbance changes recorded on a home-built spectrophotometer [27]. The P/P^+^ signal amplitude and P^+^Q_A_^−^ → PQ_A_ charge recombination kinetics were recorded at 430 nm and decomposed into exponentials. The remaining secondary quinone activity of the RC was inhibited by the addition of 100 μM terbutryn, which blocked the interquinone electron transfer. The RC concentration was determined from the steady-state optical densities at 802 nm or 865 nm, using extinction coefficients of *ε*_802_ = 0.288 µM^−1^ cm^−1^ or *ε*_865_ = 0.135 µM^−1^ cm^−1^ [27,44]. The charge recombination to the ground state may occur via direct and indirect (thermally accessed through P^+^I^−^) tunneling pathways. The observed rate constant is as follows:(1)$$k_{\mathrm{obs}}=k_{A}+k_{I}\cdot \exp(-\Delta G^{o}/k_{B}T),$$
where *k*_A_ represents the P^+^Q_A_^−^ → PQ_A_ direct charge recombination and *k*_I_ represents the quinone-independent decay rate constant of the P^+^I_A_^−^ → PI_A_ charge recombination, taken here as 7.7·10^7^ s^−1^ [47]. Δ*G*^o^ is the free energy difference between one of the relaxed states of P^+^I_A_^−^ and P^+^Q_A_^−^, *k*_B_ is the Boltzmann constant, and *T* is the absolute temperature.

The delayed fluorescence of the BChl dimer was measured by a home-built kinetic fluorometer equipped with a frequency-doubled and Q-switched Nd:YAG laser (Quantel YG 781-10, Newbury, UK, wavelength 532 nm, energy 20 mJ, duration 5 ns). The photomultiplier was protected by an electronically controlled mechanical shutter (Uniblitz VS25, Rochester, NY, USA) [48,49,50,51,52], or by the electronic switching of the dynodes [53] using an analog or time-correlated single-photon-counting signal processing, respectively. The free energy change between P* and P^+^Q_A_^−^, Δ*G*^o^, was calculated via a comparison of the delayed and prompt fluorescence yields, as follows [54]:(2)$$∆G^{o}=k_{B}T\cdot \left(\frac{\int F_{d}\cdot dt}{\int F_{p}\cdot dt}\cdot \frac{k_{d}}{k_{fl}}\cdot \frac{\eta_{fl}}{\eta_{ph}}\right)\;.$$

Here, ∫*F_d_*(*t*)*dt* and ∫*F_p_*(*t*)*dt* are the integrated intensities of the delayed and prompt fluorescence, respectively, measured in the same sample but at very different excitation intensities (both in the linear region) to provide similar emission intensities; *k_B_**T* (=25 meV at room temperature) is the Boltzmann factor; *k_fl_
*(=8·10^7^ s^−1^) is the radiative rate constant of prompt fluorescence; *k_d_* is the decay rate constant of the delayed fluorescence; *η_ph_* (≈1.0) is the quantum yield of photochemical trapping; and *η_fl_* (=4·10^−4^) is the quantum yield of the prompt fluorescence.

The temperature of the sample was controlled by a thermostat in the physiological temperature range and measured by a thermocouple to a precision of 0.3 °C [38].

### 2.4. Models of the Electrostatic Interactions

#### 2.4.1. Calculation of the pH-Dependence of the Free Energy Change Due to Q_A_/Q_A_^−^ and I_A_/I_A_^−^ Transitions

The cluster consisting of *n* closely connected acidic residues interacts with nearby (Q_B_^−^) or remote (Q_A_^−^ and I_A_^−^) redox centers, leading to the slight stabilization of the free energy in the alkaline pH region [34,36,55,56,57,58]. The protonation pattern in the acidic cluster can be calculated according to an anticooperative model (i.e., the proton binding of one residue in the cluster disfavors additional proton binding to a second residue) [59]. The residues are neutral when protonated (with binary number 0) and anionic when deprotonated (with binary number 1). The probability of a particular protonation configuration of the cluster (*k*) is given by the following:(3)$$P_{k}\left(pH\right)=\frac{10^{B_{k}\cdot (M(pH)\cdot B_{k})}}{\sum_{i=0}^{2^{n}-1}10^{B_{i}\cdot (M(pH)\cdot B_{i})}}\;\;,$$
where *B_i_* is the vector, whose elements are the binary digits of number *i*; *M*(*pH*) is the *n* × *n* matrix, whose diagonal elements are (p*H*–p*K*_i_) and nondiagonal elements are *E*_ij_/2. Here, p*K*_i_ denotes the intrinsic p*K*_a_ value of residue *i* and *E*_ij_ denotes the mutual interaction energy between charged groups *i* and *j*, expressed in units of *RT*·ln(10) ≈ 60 meV (at room temperature), which increases the intrinsic p*K*_i_ to (“dark” sate) p*K*_i_^D^. In the presence of a light-induced negative charge on the interacting redox center (Q_A_^−^, Q_B_^−^ or I_A_^−^), the p*K*_i_^D^ of the *i*-th residue in the cluster is shifted further due to the Coulombic interaction energy (*V*_i_): p*K*_i_^L^ = p*K*_i_^D^ + *V*_i_. The (right and left) scalar products with vector *B* (indicated by dots) select the interaction terms corresponding to all couples of charged groups. The probability of the protonation of a selected residue in the cluster can be determined using the calculated probabilities of all *k* configurations. The number of protonated groups in the cluster is $n-\sum_{i=1}^{n}{{((B}_{k})}_{i})$ and
(4)$$H\left(pH\right)=n-\sum_{k=0}^{2^{n}-1}P_{k}(pH)\sum_{i=1}^{n}{{((B}_{k})}_{i})\;$$
shows the proton uptake of the cluster at an arbitrary pH. The light-induced proton uptake is the difference between the protonation of the cluster in the “light” and “dark” states:(5)$$∆H\left(pH\right)=H^{L}\left(pH\right)-H^{D}\left(pH\right)\;.$$

The integrated proton uptake associated with the appearance of a negative charge on the redox center reflects the influence of the protonation on the energetics of the light–dark process, according to the following equation:(6)$$\Delta G_{H}^{o}\left(pH\right)=k_{B}T\cdot \ln\;10\cdot \int_{pH}^{pH^{0}}∆H(pH)dpH,$$
where *k*_B_*T* is the Boltzmann energy. $\Delta G_{H}^{o}$ represents the pH-dependent contribution to the free energy of the charge separation, relative to a reference pH at which the integration is started.

#### 2.4.2. Screening of the Electrostatic Interaction between Charge Pairs in the RC

(a) Single planar boundary (Figure 3A,B). The interacting point charges on Q_A_ and the acidic cluster (Q_A_ ↔ cluster, distance *R* = *R*_13_ = 14.5 Å) and on I_A_ and the cluster (I_A_ ↔ cluster, distance *R* = *R*_23_ = 25.2 Å) are in the RC (dielectric constant *D*_RC_) separated from the aqueous phase (with no electrolyte, dielectric constant *D*_w_) by a flat boundary. The polarization term can be handled using the method of image charges [60]. The solvent screening effect is incorporated into the effective dielectric constant between the two charges, as follows:(7)$$D_{\mathrm{eff}}=\frac{D_{\mathrm{RC}}}{1+\frac{R}{R^{\prime }}\cdot \frac{D_{\mathrm{RC}}-D_{w}}{D_{\mathrm{RC}}+D_{w}}}\;\;,$$
where *R*′ denotes the distance of the charge on Q_A_ or on I_A_ from the image of the charge of the cluster located at a distance *x* from the dielectric boundary. *R*′ can be expressed by *x*. The effective dielectric constant is a monotonously decreasing function of *x* from *D*_eff_ = (*D*_w_ + *D*_RC_)/2 (if *x*/*R* → 0) to *D*_eff_ = *D*_RC_ (if *x*/*R* >> 1) (Figure 3B).

(b) Sandwich planar boundaries (Figure 3C,D). The two point-charges on I_A_ and the cluster are separated by distance *R* (=25.2 Å) and are in the middle of the hydrophobic part of the RC of width *x* and dielectric constant *D*_RC_. The RC is sandwiched by the aqueous phases of dielectric constant *D*_w_ on the cytoplasmic and periplasmic sides by the protein. The interaction between the charges can be described by an infinite set of discrete image charges of alternating sign and separation *x*. The image charges are associated with one of the two point-charges so that the other point charge interacts not only with its original partner but also with all its images [61]. The effective dielectric constant is as follows:(8)$$D_{\mathrm{eff}}=\frac{D_{RC}}{1+2\cdot \sum_{n=1}^{\infty }\frac{{\left(\frac{D_{RC}-D_{w}}{D_{RC}+D_{w}}\right)}^{n}}{\sqrt{1+{\left(\frac{x}{R}\right)}^{2}\cdot n^{2}}}},$$
and describes the decrease in the interaction energy between the two charges due to the screening of the aqueous phases around the RC. The effective dielectric constant is a monotonously decreasing function of *x* from *D*_eff_ = *D*_w_ at *x* = 0 to *D*_eff_ = *D*_RC_ at *x* >> R (Figure 3D).

MathCad 14.0 was used for the numerical calculations.

## 3. Results

The kinetics of charge recombination were measured in the RCs of the purple bacterium *Rba. Sphaeroides*, where the native ubiquinone_10_ (UQ_10_) was replaced by low-potential anthraquinones at the primary quinone binding site (Figure 4 top). The charge recombination from the P^+^Q_A_^−^ state can proceed along two parallel routes, and the rate of the observed reaction will be the sum of the rates of the two reactions ([62], Equation (1)). The first reaction is a direct tunneling to the ground state, and the second one is an uphill (indirect) reaction to P^+^I_A_^−^, where P^+^Q_A_^−^ pre-equilibrates with P^+^IA^−^ before decaying to the ground state (Figure 2). Through the replacement of various quinones of very different midpoint redox potentials for the native Q_A_ (UQ_10_), P^+^Q_A_^−^ will recombine indirectly via P^+^I_A_^−^ if Δ*G** ≤ 0.8 eV, and directly if Δ*G** ≥ 0.8 eV [13]. The free energy level of the P^+^Q_A_^−^ state relative to that of the P^+^I_A_^−^ state can be determined from the relative contribution of the uphill reaction [13,48,63,64,65]. In all our cases, faster kinetics were obtained than those obtained with UQ_10_ at the Q_A_ site, indicating the contribution of the indirect pathway of the charge recombination. The lower the midpoint potential of the anthraquinone, the more significant the Boltzmann term in Equation (1); therefore, the rate constant of the back reaction increases. As the energy gap for the indirect pathway of recombination depends on the interactions with protonatable groups, the observed rate constants depend not only on the temperature but on the pH as well. The sharp difference between the pH dependence of the charge recombination rates for AQ and UQ_10_ is demonstrated in Figure 5. Because the rate of direct charge recombination is hardly controlled by the driving force, a very slight pH dependence can be observed if Q_A_ is the native ubiquinone. However, if UQ_10_ is replaced by AQ, the RC will increase the rate of the back reaction upon an increase in pH. In contrast to earlier work [66], no signs of saturation can be seen in the pH range studied here.

According to Equation (1), the free energy gap of the thermal activation Δ*G*^o^(P^+^I_A_^−^Q_A_ → P^+^I_A_Q_A_^−^) is worthy of being introduced in the place of the rate constants. The van’t Hoff analysis of the temperature dependence of the measured rate constants shows the difference in the free energy levels of P^+^I_A_^−^Q_A_ and P^+^I_A_Q_A_^−^, which is shown to be slightly pH-dependent in the alkaline pH region (Figure 6). This is a strong indication that either Q_A_^−^ or I_A_^−^ or both interact with protonatable residues in the RC and the interactions are slightly different. The van’t Hoff analysis of the rate constants of the charge recombination provides information about the pH dependence of the difference in the free energy levels but not about the pH shifts in the individual P^+^I_A_^−^Q_A_ and P^+^I_A_Q_A_^−^ states. The observed difference may derive exclusively from the stabilization of the Q_A_^−^ state at a low pH or from the stabilizations of both states to different extents. If we intend to distinguish the two effects and to determine the pH-induced displacement of the state of I_A_^−^ separately, the measured values should be corrected to those attributed to Q_A_^−^. We need a different and independent method for the separation. The delayed fluorescence of the bacteriochlorophyll dimer is an appropriate assay. Through a comparison of the yields of the prompt and delayed fluorescence of P*, the absolute values of the free energy states of the low-potential quinones can be obtained.

As the delayed fluorescence derives from the leakage of the charge-separated state P^+^Q_A_^−^, it will follow the kinetics of charge recombination (Figure 4 bottom). Because of the relatively slow (about 100 ms) back-reaction of RC with native UQ_10_, the delayed fluorescence can be measured by mechanical shutter with a good signal-to-noise ratio. However, the recombination times become much smaller (in the ms time range) upon the replacement of UQ_10_ by low-potential quinones and, accordingly, the kinetics of the delayed fluorescence will be faster. The slow response of the mechanical shutter will limit the time resolution of the kinetics. The electronic switching of the photomultiplier and time-correlated single-photon-counting of the signal were used to avoid the artifacts caused by the intense prompt relative to the very weak delayed fluorescence of the sample [50,53]. With this method, the charge recombination kinetics can be resolved at the sub-millisecond timescale and the free energy level of the P^+^Q_A_^−^ state relative to that of P*Q_A_ can be determined from the measurement of the delayed fluorescence of the BChl dimer (see Equation (2)). Figure 7 demonstrates these values for RCs where the native UQ_A_ was substituted by a series of low-potential AQ derivatives. The free energies show either no (1-chloro-AQ) or (above pH 10) slight (≈10–15 meV/pH unit) pH dependence. These values are in good agreement with those of previous measurements [13,48]. To verify that the pH dependence observed for the rates of charge recombination and the intensity of delayed fluorescence were not artifacts of the quinone removal and reconstitution process, they were also measured for RCs reconstituted with UQ_10_. The essential pH independence for *k*_AP_(UQ_10_) and slight pH dependence for ∆*G*_P*A_(UQ_10_) were in good agreement with those presented earlier [48,49,67].

If the free energy gap between P^+^I_A_^−^Q_A_ → P^+^I_A_Q_A_^−^ is subtracted from the free energy level of P^+^Q_A_^−^, the free energy state of I_A_^−^ can be obtained. Comparing the two sets of curves for different AQ substituents, the following conclusions can be drawn: (1) It is not expected that the free energy state of I_A_^−^ would depend on the chemical nature of the substituent at the Q_A_ binding site. However, the free energy levels of I_A_^−^ for different substituents are not unified into a single trace but constitute a narrow branch of curves. This indicates that the lowest level of P^+^I_A_^−^ relaxation will depend slightly on the chemical nature of Q_A_. Different Q_A_s will cause different degrees of relaxation of P^+^I_A_^−^. (2) The free energy levels of P^+^I_A_^−^ show a definite pH decrease in the highly alkaline pH range (pH > 10) to an extent comparable to that of P^+^Q_A_^−^. Similar residues of high p*K*_a_ values may participate in the pH-dependent interaction with the negative charge on Q_A_ and I_A_. Likewise, for P^+^Q_A_^−^, 1-chloro-AQ shows no pH dependence, indicating the possibility of a unique binding structure at the Q_A_ site, which may reduce (block) the electrostatic interactions with the cluster of residues.

The standard free energies between P^+^Q_A_^−^/P* and P^+^Q_A_^−^/P^+^I_A_^−^ obtained from direct measurements of the intensity of the DL and the rates of charge recombination consist of enthalpic (Δ*H*^o^) and entropic (*T*·Δ*S*^o^) terms, Δ*G*^o^ = Δ*H*^o^ − *T*·Δ*S*^o^, which can be determined from temperature-change measurements [38]. The enthalpy changes can be obtained from the temperature dependence of the amplitude (integral) of the decay of the delayed fluorescence as the slopes of the straight lines show the enthalpy change in the charge separation (van’t Hoff plot, Figure 8). The thermodynamic parameters for the different quinones in the Q_A_ site are summarized in Table 1. The changes in the free energy and the enthalpy of the P* → P^+^Q_A_^−^ and P^+^I_A_^−^ → P^+^Q_A_^−^ transitions are negative for all quinone analogs, indicating spontaneous and exothermic reactions, respectively. The entropy changes show large variations upon quinone substitution and are positive (with the assumption of 1-chloro-AQ in the P^+^I_A_^−^ → P^+^Q_A_^−^ transition), representing an increase in the disorder of the system.

## 4. Discussion

The bacterial RC presents exceptional opportunities for studying charge compensation and conformational relaxation in proteins. Immediately after charge separation, the RC begins to reorganize around the newly formed anion and cation via protein relaxation [68,69] and/or the static distribution of conformational heterogeneity [70], resulting in a time-dependent decrease in the standard free energy of the P^+^I_A_^−^ (Figure 2). Additionally, our results demonstrated the role of protonation in the early steps of the charge stabilization, which is the focus of our discussion.

A novel aspect of our investigation was the experimental determination of the drop in the free energy level of the P^+^I_A_^−^ state in the alkaline pH range, which displayed similarities with those of the P^+^Q_A_^−^ and P^+^Q_B_^−^ semiquinone states in terms of both (1) pH dependence and (2) magnitude. (1) This observation suggests a common origin of the interaction of P^+^I_A_^−^ with the acidic cluster in the Q_B_-binding domain. (2) Compared to the semiquinones, the interaction with P^+^I_A_^−^ could be more complex due to the hydrophobic location and to the short lifetime of the P^+^I_A_^−^state, which may increase the kinetic constraint of the internal protonation processes in the cluster. However, the replacement of the native UQ_A_ with low-potential quinones increases the lifetime of P^+^I_A_^−^ by about five orders of magnitude (from 10 ns to 1 ms) at the expense of the corresponding decrease in its apparent concentration. The electrostatic and kinetic control of the interaction between P^+^I_A_^−^ and the acidic cluster around Q_B_ will be surveyed.

### 4.1. Interaction of the P^+^I_A_^−^ State with the Acidic Cluster at Q_B_

The RC shows a similar H^+^ pattern (pH dependence of the free energy states) in response to the establishment of Q_A_^−^, Q_B_^−^, and I^−^, which means that many of the same groups experience the bulk of the conformational changes and p*K*_a_ shifts despite the different locations of the charge and by the fact that all are closer to the Q_B_ site. This is because the local hydrophobic dielectric around Q_A_^−^ and I_A_*^−^* is rather ineffective in screening the negative charges, which have a long-range electrostatic influence [71]. However, the charges in the acidic cluster might be effectively screened by the mainly hydrophilic character of the environment. The X-ray structure of the RC shows several (up to six) water molecules in the Q_B_ site when Q_B_ is absent [41,72,73], suggesting the invasion of water molecules into the Q_B_ site if it is not occupied by the ubiquinone. This was the case in our experiments, as the Q_B_ activity of the RC was not restored after UQ_A_ was replaced by quinone analogs; therefore, the charges in the acidic cluster near the Q_B_ binding site should be screened via the invading water molecules. The electrostatic pattern is further complicated by a loose cluster of water molecules extending almost from the Q_A_ site to the Q_B_ site. The water molecules in the cluster are sufficiently ordered to be well defined in the X-ray structure [74], and the linkage (denoted as “wire”) between the two quinone sites was used to understand the modification of flash-induced proton binding in various mutants of the RC [33,75]. The chain of water molecules might function as a polarizable transmitter of the electric potential of Q_A_^−^ to the acidic cluster. Studies on charge–charge interactions between P^+^ and the ionizable amino acid residues introduced by mutations at selected sites suggest that they allow for a strong electrostatic screening [76]. Counterions make major contributions to this screening, but the penetration of water molecules or other relaxation processes could also play a substantial role.

Despite the highly complex nature of the electrostatic interactions, we tried to apply a minimum electrostatic model to obtain quantitative support for the observed phenomena.

#### 4.1.1. Comparison of the Measured and Calculated Free Energies of Q_A_/Q_A_^−^ and I_A_/I_A_^−^ at a High pH

The observed slight increase in the free energies at a high pH (Figure 7) can be well described by the anticooperative proton binding model of the acidic cluster ([59] as shown in M&M) using reasonable assumptions. (1) The intrinsic p*K*_a_ values of all acidic residues in the cluster (Asp and Glu) were taken as their solutional values of 4.5. (2) The interaction energies among all couples (within and outside the cluster) are inversely proportional to their distances. (3) The pair energies in the cluster are not affected by the substitution of the quinone analogs to the Q_A_ binding site. Figure 9 demonstrates how well the calculated model with four amino acids in the cluster approaches the measured points when the native ubiquinone and 1-amino-AQ are on the Q_A_ binding site. The data for all quinone substitutes used in this study are collected in Table 2.

Similarly to spectroscopic [77] and FTIR [40] studies on site-specific mutants, our results identified GluL212 (in interaction with AspL213) as the major contributor in this process. These two acidic residues have different levels of accessibility to the protein surface, which determines which is ionized first. GluL212 is the residue that binds a proton when Q_B_ is reduced [40,78]. As AspL213 is ionized at a lower pH, it suppresses the ionization of GluL212, resulting in an unusually high p*K*_a_ (≈10). Since the net charge on the two acidic residues remains the same, with only one being ionized up to pH 9, the precise distribution of the cluster protonation has only modest effects on the interaction with surrounding groups [79]. The closely spaced and strongly interacting residues in the cluster result in a decrease in the slope of the high pH dependence of the free energy change and the pH titration deviates markedly from the classical Henderson–Hasselbalch titration.

The observed interaction energies of the cluster with the light-induced Q_A_^−^ and I_A_^−^ anions showed a strong dependence on the chemical structure of the quinone analog at the Q_A_ site (Table 2). This can be considered as manifestation of the cross-talk between the sites. The signaling of structural information introduced at the Q_A_ binding site occurs from one site to the other via a long-distance influence on the ionization and conformational states of the cluster. Isoprenyl ubiquinones induced a relatively large protonation/conformational configuration [64]. In contrast, the planar naphthoquinones allowed for a small structural relaxation only. Additionally, some mutations of RCs where the naphthoquinones were used as Q_A_ appeared to “break” the linkage between the two quinone sites. The most notable of these were mutations of ProL209 to Phe, Tyr, and Trp [75], which were characterized by X-ray diffraction analysis. Similarly, a great reduction in, and even elimination of, the high-pH proton uptake to the Q_A_^−^ state was observed when certain naphthoquinones (e.g., menadione (2-methylnaphtoquinone)) were substituted for the native ubiquinone (Q-10) in the Q_A_ site [46]. Similarly, the lack of protonation/deprotonation of the cluster in the alkaline pH range was observed in this study when 1-chloro-AQ occupied the Q_A_ binding site. The elimination of the pH dependence of the free energy of both Q_A_^−^ and I_A_^−^ redox states indicated that the 1-chloro-AQ at Q_A_ site broke the (dynamic and/or energetic) cross-talk with the acidic cluster.

All these observations reflect the connection between the Q_A_ and I_A_ binding sites and the remote acidic cluster, resulting in a structurally sensitive communication. The interaction can be mediated by possible contact between the redox centers and essential amino acids in the binding pockets (e.g., methionine M218 [49] and tryptophane M252) and by the extended network of internal cavities in the RC. Thus, the perturbation caused by quinone replacement can spread out to the I_A_ binding pocket and to the acidic cluster via histidine M219 and the His–iron complex. The result indicates that the energy landscape is both complex and subtle and requires a higher experimental resolution than is currently available, as well as a correlation with more structurally informative methods (e.g., mapping surface cavities using CASTp; see below) [80].

#### 4.1.2. Role of Protein in Determination of the Interaction Energies

There are several indications that the electrostatic interaction between charges in the RC is not governed by simple electrostatics in homogeneous media (where the energy of the point charge is proportional to 1/(*D·r*)); instead, the picture is shaded by local polarizations. It was revealed that the acidic cluster had a similar interaction energy to I_A_^−^ and Q_A_^−^ (Table 2), although they were in different environments with highly different dielectric constants and at comparable distances from the cluster (Figure 1). The problem is similar when the free energies of the Q_A_^−^ and Q_B_^−^ states are compared: although the distances to the cluster are significantly different, the free energy gap is even smaller (60 meV at a neutral pH, which gradually disappears with an increase in pH at pH ≈11). Following the introduction of a charge to Q_A_, Q_B_, or I_A_, the RC provides delocalized responses, which are coordinated by an interactive network that links the electron acceptors, polarizable and protonatable residues, and numerous water molecules that are located at the cytoplasmic region of the RC [27,33].

Simple models of single- (Figure 3A) and double-planar interfaces (Figure 3C) between the hydrophobic and aqueous phases were used to consider the effects of polarization on the effective dielectric constant in analytical forms. Although the models were highly simplified and served as a minimum form of the actual arrangement, they offered a quantitative approximation of the variation in the effective dielectric constant in the vicinity of the aqueous phase. Taking reasonable values for the distances (Q_A_ from the aqueous phase *x* ≈ 5 Å (model A) [66] and the thickness of the hydrophobic belt of the RC *x* ≈ 42 Å (model C) PDB ID: 3I4D), *D*_eff_ = 10.0 ± 0.5 and *D*_eff_ = 6.5 ± 0.5, as effective dielectric constants for the Q_A_ ↔ cluster and I_A_↔cluster’s electrostatic interactions can be represented. These values are in good agreement with the estimates from [71]. As the products of *D*_eff_·*R* are very close in the two cases, the interaction energies should be similar, in good accordance with the experimental results (Figure 7 and Figure 9). The hydrophobic region around Q_A_ decreases, and the vicinity of the aqueous phase around I_A_ increases, the effective dielectric constant to keep the Q_A_↔cluster and I_A_↔cluster electrostatic interactions at similar levels.

A closer look at the structure of the RC will support this conclusion. The isolated (membrane-free) RC protein is not tightly packed by the cofactors and amino acids but has extended cavities that are accessible for solute molecules. Remarkable cavities can be observed in the hydrophobic belt of the RC, connecting the Q_A_ binding pocket with those of the two bacteriopheophytins and the Q_B_ (Appendix A). Part of the porphyrin ring of I_A_ is at the entrance of the surface cavity, and is thus exposed to the solute (Figure 10).

The arrangement provides an opportunity for the electrostatic screening of the charge on I_A_^−^ in aqueous solution, resulting in increases in the effective dielectric constant, as shown in the simplified electrostatic models.

#### 4.1.3. Dielectric Relaxation of the Free Energy of the P^+^I_A_^−^ State

When the native ubiquinone is substituted by quinones with substantially more negative midpoint potentials, the decay of P^+^Q_A_^−^ proceeds via a thermally accessible intermediate state (Figure 2). A long-standing question regards the nature of the intermediate state. As the energetics of the relaxation do not depend strongly on the nature of the substituted quinones, the intermediate state cannot be an activated form of P^+^Q_A_^−^. According to the electrostatic interaction energy calculations in *Blastochloris viridis*, the P^+^I_A_^−^ radical pair lies at about 87 meV, below the lowest excited singlet state of the dimer (P*), when the radical pair is formed in the static crystallographic structure [81]. The reorganization energy for the subsequent relaxation of P^+^I_A_^−^ was calculated to be 217 meV, so the relaxed radical pair lies about 304 meV below P*. Here, we observed a somewhat larger stabilization energy for the P^+^I_A_^−^ charge-separated state of RCs from *Rba. sphaeroides*. The dielectric relaxation is deeper and depends slightly on the chemical nature of the substituting quinone: the average free energy level is −370 ± 20 meV. The drop in the free energy of the slow dielectric relaxation relative to P* is comparable to the free energy level of the (unrelaxed) state of the P^+^I_A_^−^ radical pair right after its formation (~210 meV, [12]). It can be concluded that the transient state is not identical to an activated form of P^+^Q_A_^−^, nor is it a hot P^+^I_A_^−^ state (P^F^) identified at the nanosecond timescale from fluorescence measurements, but a highly relaxed form of the P^+^I_A_^−^ state formed via a substantial (≈(370 ± 20) meV) relaxation after the initial charge separation at the long (millisecond) timescale.

Our experiments clearly showed that a small part of this substantial dielectric relaxation was pH-dependent (Figure 7). In the alkaline pH range above pH 10, a slight increase in the free energy of the P^+^I_A_^−^ state was observed, which was attributed to the interaction with the distal acidic cluster at around Q_B_. This pH-dependent interaction energy adds up to non-negligible part of the total relaxation energy of 370 meV. Due to the lack of reliable measured data at extremely high (>11.5) pH values, the maximum energy attributed to protonation cannot be obtained experimentally during the initial rise, nor the plateau can be measured in the very alkaline pH range. We can make a lower estimate of 60–80 meV, which means that the (full) protonation includes a stabilization energy of about 20% of the total relaxation energy. This is roughly the same magnitude as that observed for the stabilization of the P^+^Q_A_^−^ charge pair (Figure 7).

### 4.2. Thermodynamics of P^+^Q_A_^−^ and P^+^I_A_^−^ Formation

The driving force of the chemical reaction is the Gibbs free energy, which is composed of enthalpic and entropic components, and their ratio has been always a matter of debate in photosynthetic systems. Although the Marcus theory of electron transfer omits the entropic contribution, recent experimental results pointed to the crucial role of the entropy change [38,47,82]. The detection of the delayed fluorescence and indirect charge recombination (Eqs. (1) and (2)) suggested a direct change in the free energy between the P* → P^+^Q_A_^−^ and I_A_^−^Q_A_ → I_A_Q_A_^−^ transitions, meaning that the changes in the enthalpy and entropy could be derived straightforwardly (Table 1). The reactions are driven mostly by the enthalpy changes, and the entropy change plays only a minor role. However, the estimation of the entropy change is important because it provides information about the possible structural changes caused by the insertion of quinone derivatives into the Q_A_ binding site.

In native RC (UQ_10_ is at the Q_A_ binding site), the formation of P^+^Q_A_^−^ appears to involve a small (≈9% of free energy change) positive entropy change, suggesting that the increase in disorder is due to the charge separation rather than due to the mobility of Q_A_. It remained in the same orientation upon illumination according to the ENDOR experiments [83] and the structure of I_A_ was found to be very similar to the structure of BPhe *a* in the solution [84]. The hydrogen bonds to Q_A_ were significantly shorter in the Q_A_^•−^ state compared to the neutral quinone state, leading to the stabilization of the radical anion [85]. A similar shortening effect was seen in the H-bond distances of the pathway to Q_B_ upon the reduction in Q_B_. The cofactors I_A_ and Q_A_ appear to be electronically close to their free forms as there is no evidence for packing effects from the protein matrix and they are not tuned in a special manner by their surroundings. The overall architecture of the Q_A_ binding site is designed to provide structural rigidity. The protein framework (Trp M252 and Ile M265 in van der Waals contact) and the H-bonds (with Ala M260 and His M219) limit the mobility of Q_A_, holding it in a position optimized for electron transfer to and from the quinone.

The reconstitution includes radical steps involving opening the pocket with a high concentration of ionic detergent followed by the insertion of a one-ring ubiquinone via a three-ring AQ in the binding pocket. These treatments may lead to various changes in the structure, resulting in changes in the entropy. While the tighter packing and loss of the isoprenoid side chain decrease the observed entropy of the system, the possible increased flexibility of the pocket, including changes in the conformation of the backbone and residues, and looser binding and heterogeneity in the orientation of Q_A_, enhance the entropy (Appendix A). Depending on the actual modifications, the observed entropy changes may show large variations for both transitions. They are mostly small but can display deviations in both directions. A small entropy change means that the electron transfer is not accompanied by essential rearrangements of the structure and the two electron sinks serving as redox centers operate under optimal conditions. However, the contribution of the entropic change to the free energy change is very large (≈65%) in RC reconstituted by 2,3 dimethyl-AQ. Both transitions are driven almost entropically when 2,3-dimethyl-AQ occupies the Q_A_ site. On the other hand, the substitution of UQ_A_ with 1-chloro-AQ decreases the disorder of the I_A_^−^Q_A_ → I_A_Q_A_^−^ reaction (*T*·Δ*S* is negative), indicating the loss (freezing) of the dynamics of the connection between I_A_ and Q_A_. Similarly unusual consequences of the 1-chloro-AQ replacement were experienced when the cross-talk between Q_A_ (or I_A_) and the acidic cluster was blocked (see above). Unfortunately, generally accepted (in PDB deposited) structural data that would support these possible changes are not available [86].

### 4.3. Comparison of the Rates of Protonation and Electron Transfer Back-Reaction

The charge recombination and the proton binding/unbinding reactions were combined as represented in Figure 7 and Appendix A. We observed that the electron back-reaction from the charge-separated state to the ground state followed the single exponential decay kinetics in all cases of quinone substitutions. This proves that the protonated/unprotonated P^+^I_A_^−^ and P^+^Q_A_^−^ states are in equilibria and the rate constants of protonation *k*_on_^H^ +*k*_off_^H^ are much larger than those of the charge back-reaction *k*_back_ ~10^3^ s^−1^.

In photosynthetic RC, the degree of the utilization of light energy for photosynthetic purposes is determined by the efficiency of the electron transfer from I_A_ to Q_A_. The large ratio of the protonation rates to those of backward recombination suggests the high efficiency of the free energy storage. In the forward process (electron transfer from I_A_ to Q_A_), the reorganization energy (including protonation) and the free energy gap are finely tuned, meaning that the process does not require activation. The matching is due to the low dielectric constant of the RC protein core because the low dielectrics strongly affect the electrostatic polarization components of both the reorganization energy and the equilibrium free energy of the reaction [71,87]. In the back reaction, however, these quantities are mismatched and lead to a high activation energy. If the protein and membrane were replaced by a homogeneous medium with a high dielectric constant, the effective energy storage would be negligible.

The method of the substitution of quinone analogs at the Q_A_ binding site used in this study offers unique advantages for the comparison of I_A_- and Q_A_-related electron and proton transfer reactions. As the lifetime of the reduced I_A_ is very short in the presence (≈200 ps) or absence (≈14 ns) of the pre-reduced form (≈8 ns) of native Q_A_ [88], the accomplishment of protonation processes in the Q_B_ cluster is not facilitated. However, the lifetime can be increased significantly (by several orders of magnitude) by replacing the native ubiquinone at Q_A_ with high-potential quinone, which enables the detection of the interaction with the acidic cluster around Q_B_. The cost of this is a similarly large decrease in the amount of I_A_^−^ participating in the interaction. An additional advantage of the replacement is the simultaneous and direct detection of the interactions of Q_A_^−^ and I_A_^−^ with the acidic cluster through the omission of the complexity originating from the different kinetics of the protonation and relaxation of the RC states.

## 5. Summary

Bacterial RC is an ideal protein to study protonation-dependent redox reactions since the electron and proton transfer to Q_B_ are closely coupled. Through a combination of different experimental approaches, such as charge recombination and delayed fluorescence based on the replacement of the native ubiquinone with high-potential anthraquinones at the Q_A_ binding site, the interaction between the acidic cluster around the Q_B_ binding site and the negative charge on either Q_A_ or I_A_ was revealed. Thus, new aspects of the coupling of the proton and electron transfer reactions based on remote interactions were discovered. These studies could allow for a deeper understanding of the mechanisms of electron transfer and proton-coupled electron transfer in other chemical and biological systems to be obtained.

## Figures and Tables

**Figure 1 biomolecules-14-01367-f001:**
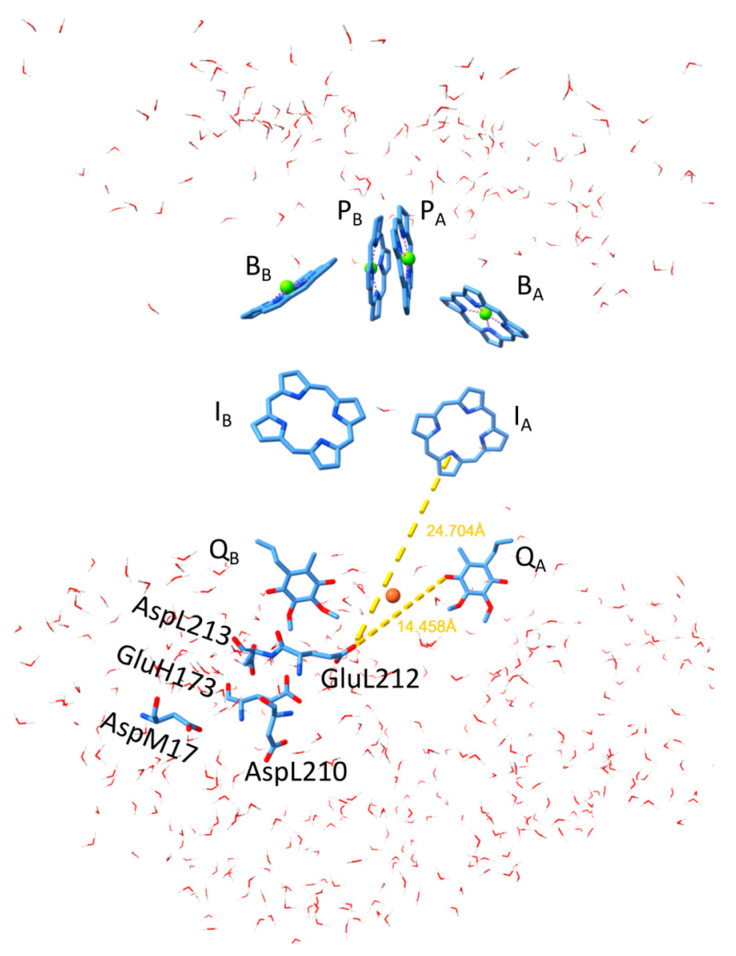
Structural view of the hydrophobic belt of the isolated RC from anoxygenic photosynthetic bacterium *Rhodobacter sphaeroides* sandwiched by aqueous phases (water molecules) at the cytoplasmic and periplasmic sides. The pairs of cofactors BChl dimer (P), monomeric BChls (B), bacteriopheophytins (I), and quinones (Q)) are arranged in active (A) and passive (B) branches. The protonation of the acidic cluster around Q_B_ plays a crucial role in the stabilization of light-induced anions in the RC. The distances of I_A_ and Q_A_ from the key residue in the cluster GluL212 are indicated. ChimeraX was used to visualize the RC of the model organism *Rb. sphaeroides* (PDB ID: 3I4D) [6].

**Figure 2 biomolecules-14-01367-f002:**
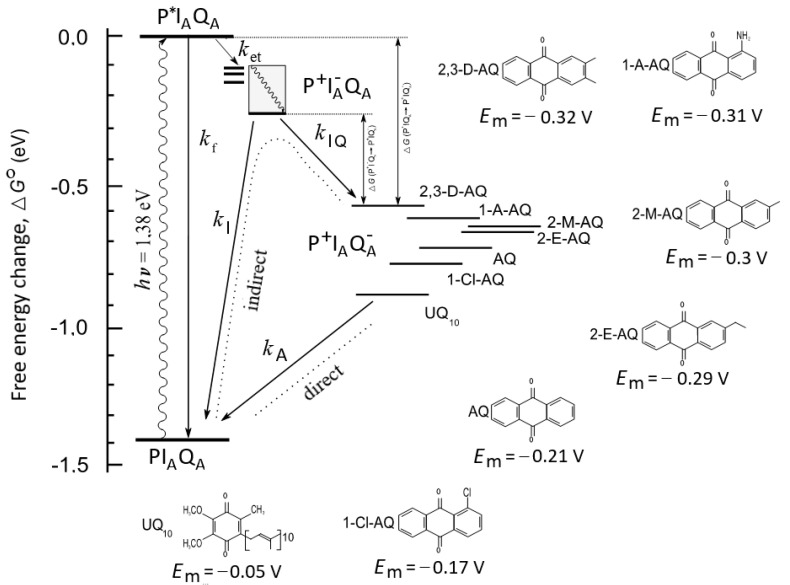
The standard free energy levels of ground (PI_A_Q_A_) and charge-separated (P^+^I_A_^−^ and P^+^Q_A_^−^) states, referring to that of the photoexcited singlet BChl dimer (P*), and the possible transitions between different states in the RC from purple photosynthetic bacterium *Rba.*
*sphaeroides*. The native ubiquinone (UQ_10_) at the primary quinone binding site is substituted by a series of low-potential quinones whose chemical structures and in situ midpoint electrochemical potentials (*E*_m_) at pH 8.0 are demonstrated. The dimer P is photoexcited (wavy arrow) and the main path of P* decay is electron transfer (*k*_et_) to Q_A_ via a transient reduction in I_A_. The charge pair can be stabilized by conformational heterogeneity (closely spaced thick lines), protein relaxation (shaded area), and proton uptake. The separated charges of the P^+^I_A_Q_A_^−^ state can be either recombined through direct or indirect pathways or can repopulate P*, which can decay through fluorescence emissions with the intrinsic rate constant *k*_f_ (delayed fluorescence).

**Figure 3 biomolecules-14-01367-f003:**
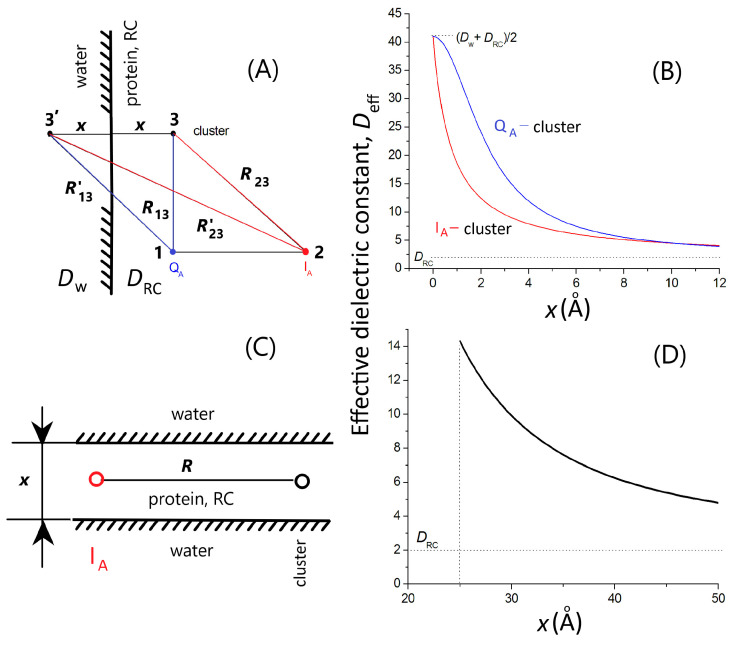
Models and demonstration of the drop in the effective dielectric constant between interactive groups (Q_A_ (1), I_A_ (2) and the acidic cluster (3)) in the RC upon an increase in the characteristic distance (*x*) from the aqueous phase. The dielectric (water) interface is represented either by a single planar boundary (**A**) or by a sandwich-type double parallel sheet (**C**). Panels (**B**,**D**) show the calculated results of models A and C, respectively. The variable *x* denotes either the distance of Q_A_ and the cluster from the dielectric (water) interface (model A) or the width of the RC between the parallel boundaries (model C). Numerical values: *R*_13_ = 14.5 Å (distance between Q_A_ and GluL212, PDB ID: 3I4D) and *R*_23_ = 25.2 Å (distance between I_A_ and GluL212, PDB ID: 3I4D), and the dielectric constants of water (*D*_w_) and the RC protein (*D*_RC_) are 80 and 2, respectively.

**Figure 4 biomolecules-14-01367-f004:**
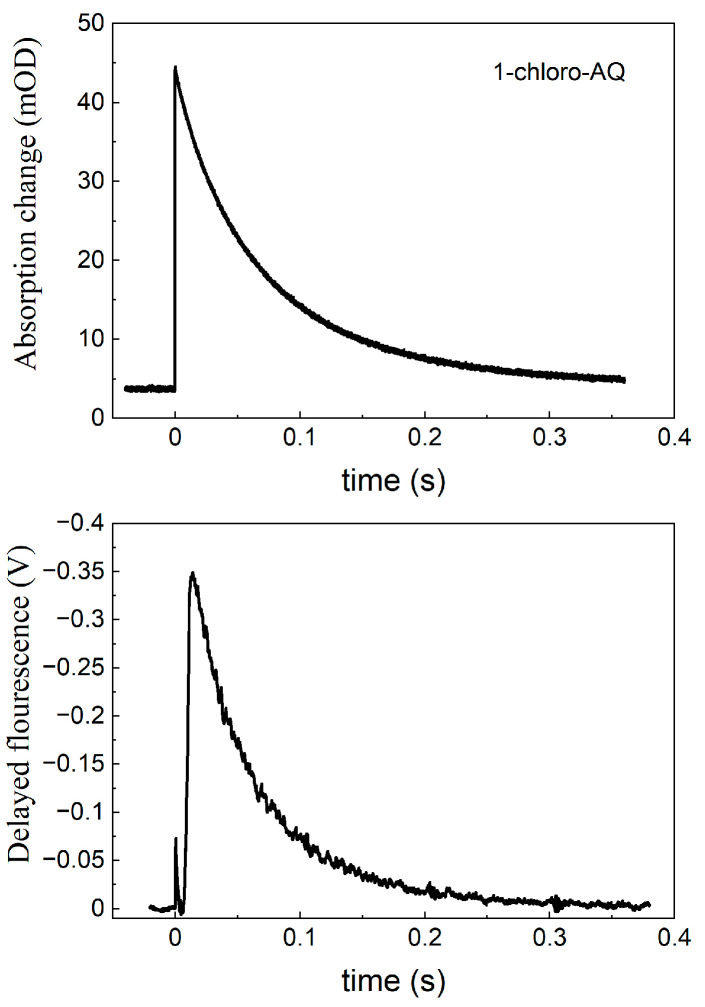
Kinetics of flash-induced absorption change (at 430 nm, **top**) and delayed fluorescence (at 910 nm, **bottom**) of RC (concentration 1.5 μM) where the native ubiquinone at Q_A_ is replaced by 1-chloro-AQ.

**Figure 5 biomolecules-14-01367-f005:**
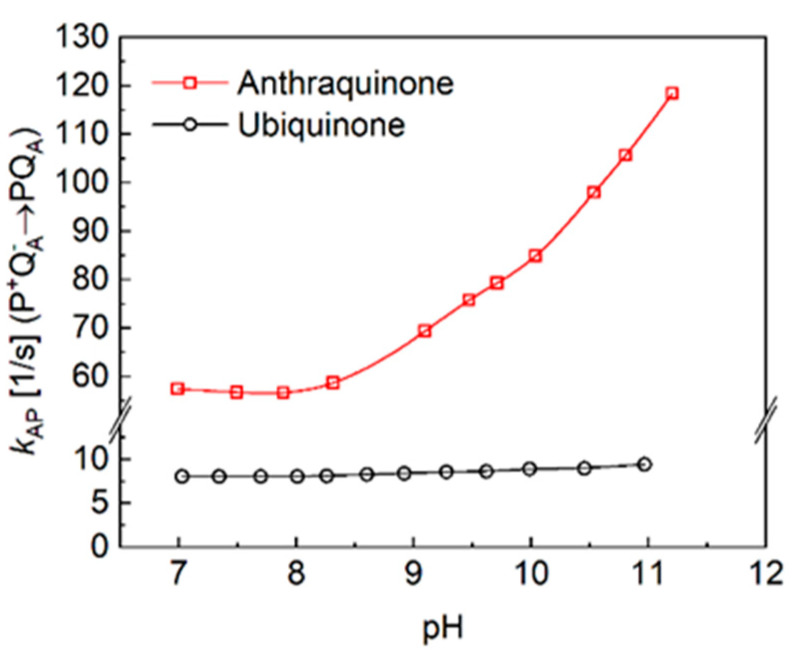
The pH dependence of the rate constant of the P^+^Q_A_^−^ → PQ_A_ charge recombination of RCs with native quinone (ubiquinone) (**bottom**) and anthraquinone (**top**) at the Q_A_ binding site. Conditions are the same as in Figure 4, except for the varying pH and buffers given in M&M.

**Figure 6 biomolecules-14-01367-f006:**
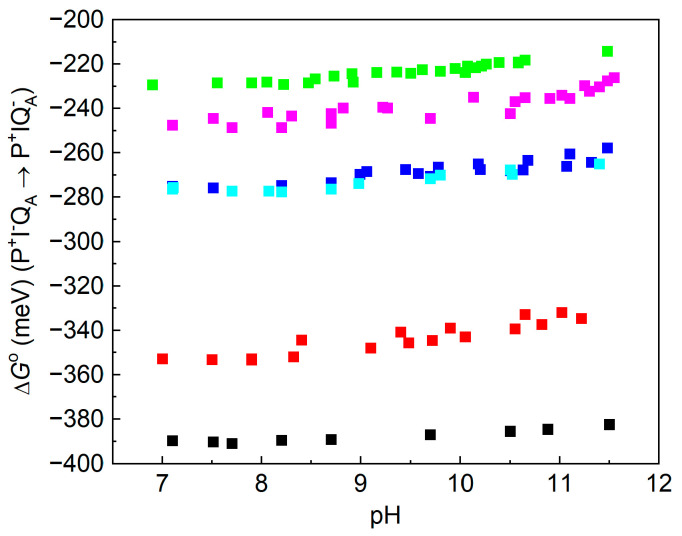
pH-dependence of the free energy states of P^+^IQ_A_^−^ relative to that of P^+^I_A_^−^Q_A_ determined from the rate constants of the temperature-dependent P^+^I_A_Q_A_^−^ → PQ_A_ indirect charge recombination kinetics via P^+^I_A_^−^Q_A_. The native UQ_10_ at Q_A_ was replaced by derivatives of the low-potential AQ: ■ 2,3-dimethyl-AQ; ■ 1-amino-AQ; ■ 2-methyl-AQ; ■ 2-ethyl-AQ; ■ AQ; ■ 1-chloro-AQ; Conditions: 1 μM RC, 0.4 mM buffer (Mes, Tris, Caps, Ches, Mops), 0.03% LDAO, and 100 mM NaCl.

**Figure 7 biomolecules-14-01367-f007:**
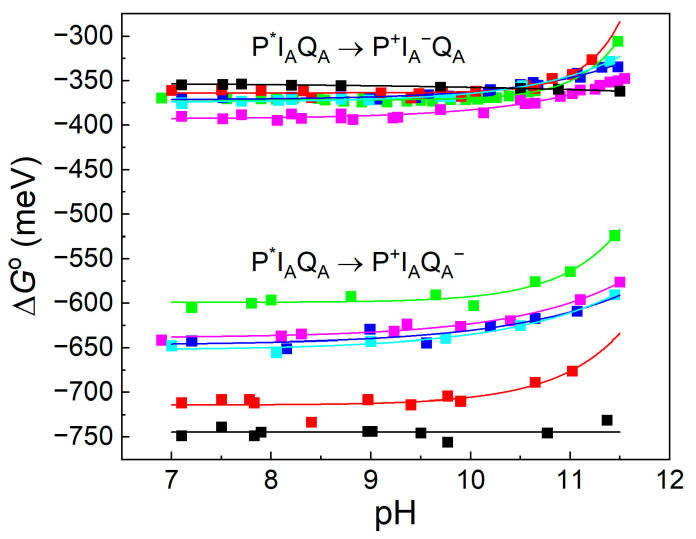
pH dependence of the free energy states of P^+^Q_A_^−^ relative to that of P*Q_A_ determined from the measurement of the delayed fluorescence of the BChl dimer (set of lower traces) and the pH dependence of the free energy states of P^+^I_A_^−^ relative to that of P*I_A_ determined from the difference in the lower traces and data in Figure 3: Δ*G*^o^(P*I_A_ → P^+^I_A_^−^) = Δ*G*^o^(P*Q_A_ → P^+^Q_A_^−^) − Δ*G*^o^(P^+^I_A_^−^Q_A_ → P^+^I_A_Q_A_^−^) (set of upper traces) in the RCs of substituted different AQ derivatives at the Q_A_ binding site. ■ 2,3-dimethyl-AQ; ■ 1-amino-AQ; ■ 2-methyl-AQ; ■ 2-ethyl-AQ; ■ AQ; ■ 1-chloro-AQ; Conditions are the same as in Figure 3.

**Figure 8 biomolecules-14-01367-f008:**
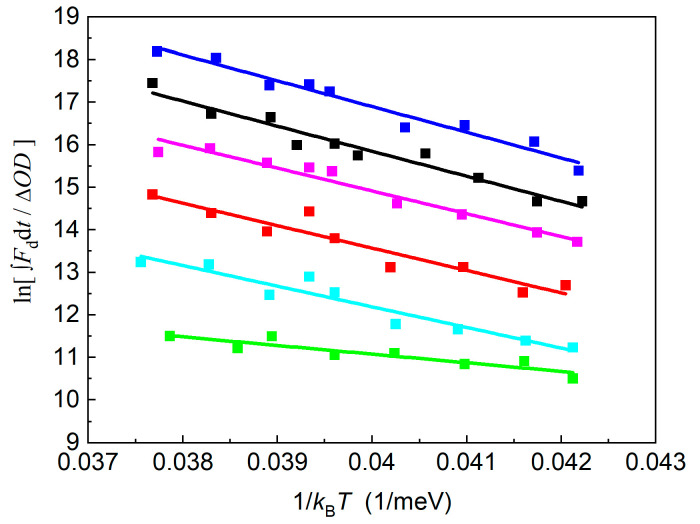
Temperature dependence of the delayed fluorescence of the BChl dimer (van’t Hoff plot) in RCs where the native UQ_10_ at the Q_A_ binding site is replaced by different low-potential forms of AQ: ■ 2-methyl-AQ; ■ 1-chloro-AQ; ■ 1-amino-AQ; ■ AQ; ■ 2-ethyl-AQ; ■ 2,3-dimethyl-AQ.

**Figure 9 biomolecules-14-01367-f009:**
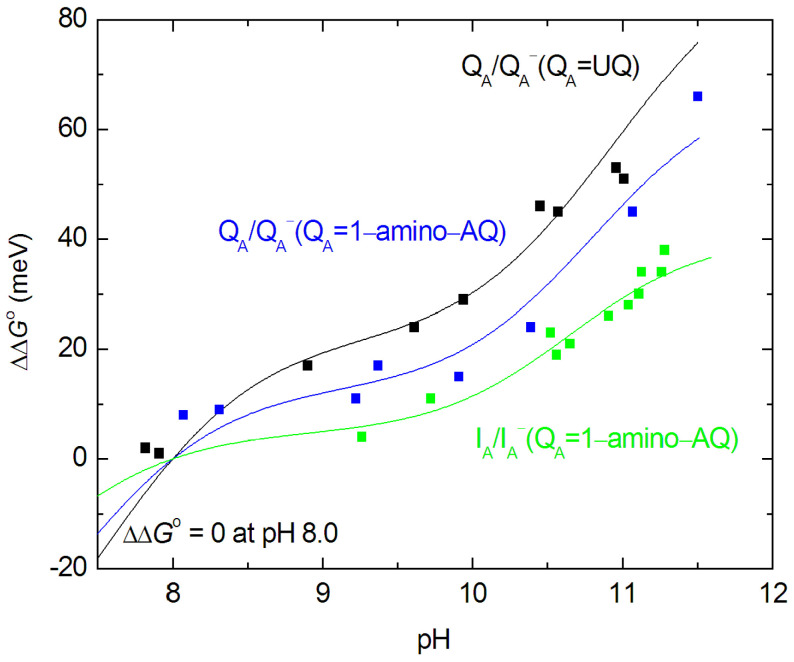
High pH dependence of the measured (data points from Figure 7) and calculated (solid lines, M&M) free energies of Q_A_/Q_A_^−^ and I_A_/I_A_^−^, referring to those at pH 8 when Q_A_ is the native UQ (black) or is substituted by 1-amino-AQ (blue for Q_A_/Q_A_^−^ and green for I_A_/I_A_^−^). The parameters of the calculated curves are given in Table 2.

**Figure 10 biomolecules-14-01367-f010:**
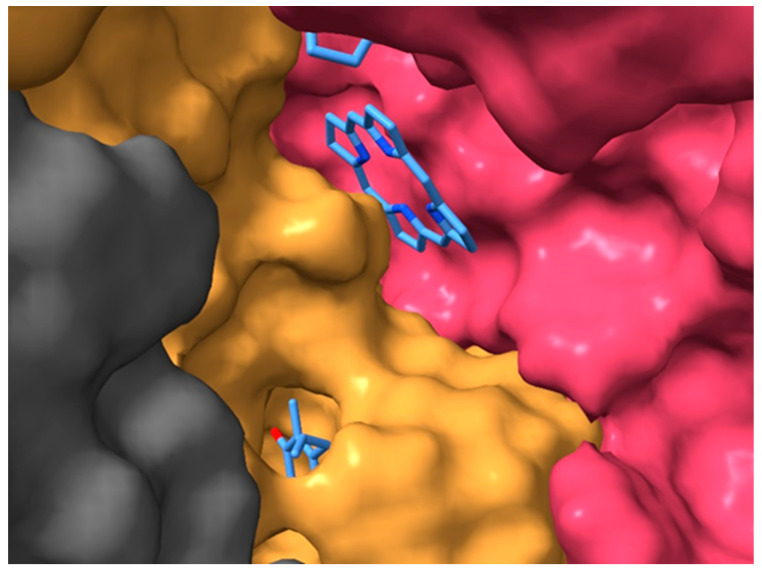
Surface exposure of I_A_ and its connection to the Q_A_ pocket. The solvent-accessible cavities of the RC of *Rba. sphaeroides* (PDB ID: 3I4D) were visualized with ChimeraX [6]. For clarity, the pythol tail of the I_A_ and the isoprenyl subunits of the Q_A_ are not shown.

**Table 1 biomolecules-14-01367-t001:** Thermodynamic (enthalpy, ∆*H*^o^ and entropy, *T*∙∆*S*^o^) costs of P* → P^+^Q_A_^−^ and I_A_^−^Q_A_ → I_A_Q_A_^−^ transitions in RCs substituted by different low-potential AQs at the primary quinone binding site Q_A_. The thermodynamic parameters of the P* → P^+^Q_A_^−^ transition were determined from the temperature dependence of the free energy difference in the states (see Figure 7) and the van’t Hoff plots of the delayed fluorescence (Figure 8). The thermodynamic characteristics of the I_A_^−^Q_A_ → I_A_Q_A_^−^ reaction were derived from the temperature dependence of the rate constants of the charge recombination (Figure 6). The temperature changed in the physiological range between 2 °C and 40 °C. The data for native UQ_10_ were taken from [48].

Q_A_	P* → P^+^Q_A_^−^			I_A_^−^Q_A_ → I_A_Q_A_^−^		
	∆*G*^o^ (meV)	∆*H*^o^ (meV)	*T*∙∆*S*^o^ (meV)	∆*G*^o^ (meV)	∆*H*^o^ (meV)	*T*∙∆*S*^o^ (meV)
UQ_10_	−910 ± 45	−830 ± 40	+80 ± 8			
2-Methyl-AQ	−625 ± 30	−605 ± 35	+20 ± 2	−357 ± 18	−330 ± 16	+27 ± 3
1-Amino-AQ	−625 ± 30	−535 ± 30	+90 ± 10	−390 ± 20	−273 ± 14	+17 ± 2
AQ	−700 ± 35	−585 ± 35	+115 ± 12	−360 ± 18	−267 ± 14	+93 ± 10
1-Chloro-AQ	−750 ± 40	−600 ± 35	+150 ± 15	−363 ± 18	−410 ± 20	−55 ± 6
2-Ethyl-AQ	−635 ± 35	−485 ± 25	+150 ± 15	−368 ± 18	−188 ± 10	+180 ± 18
2,3-Dimethyl-AQ	−585 ± 30	−205 ± 15	+380 ± 40	−365 ± 18	−72 ± 4	+293 ± 30

**Table 2 biomolecules-14-01367-t002:** Distances (in parentheses, measured in Å) and interaction energy (in meV) within the amino acids of the cluster and between the cluster and the negative charges on Q_A_ and I_A_ calculated using the fit of the model in M&M to the high measured pH dependence of the free energy change.

		GluL212	AspL213	AspM17	GluH173
GluL212		–	(9.65)	(13.81)	(8.56)
AspL213		−168	–	(8.46)	(10.0)
AspM17		−118	−192	–	(9.28)
GluH173		−190	−162	−174	–
UQ_10_	Q_A_	(14.46)	(22.42)	(25.69)	(16.74)
		−96	−61	−54	−83
	BPheo_A_	(25.20)	(30.91)	(35.63)	(27.11)
		–	–	–	–
1-amino-AQ	Q_A_	−75	−48	−42	−65
	BPheo_A_	−41	−34	−30	−38
2–3 dimethyl-AQ	Q_A_	−84	−54	−47	−72
	BPheo_A_	−14	−11	−10	−13
AQ	Q_A_	−48	−31	−27	−41
	BPheo_A_	−45	−37	−32	−41
2-methyl-AQ	Q_A_	−84	−54	−47	−72
	BPheo_A_	−55	−45	−39	−51
2-ethyl-AQ	Q_A_	−87	−56	−49	−75
	BPheo_A_	−53	−44	−38	−49

## Data Availability

The raw data supporting the conclusions of this article will be made available by the authors upon request.

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
