# Peer review of "Contribution of Protonation to the Dielectric Relaxation Arising from Bacteriopheophytin Reductions in the Photosynthetic Reaction Centers of Rhodobacter sphaeroides"

_biomolecules, 2024, doi:10.3390/biom14111367_

Round 1
Reviewer 1 Report
Comments and Suggestions for Authors
The paper "Protonation is part of dielectric relaxation upon reduction of bacteriopheophytin in reaction center of photosynthetic bacterium Rhodobacter sphaeroides." by Sipka and Maroti focus on the assessment of the contribution of the protonation state of the primary electron acceptor QA to the dielectric relaxation of BPheo(A). The manuscript is well written and very insightful. The experiments are well organised and the models are highly adherent to the physics on the electron/proton transfers within the bacterial reaction center.
Overall, the manuscript is of high quality and deserves to be published as it is. The introduction paragraph is a bit lenghty, but very informative and rich of useful literature.
Author Response
Many thanks for the positive outcome of the evaluation of the MS.
I appreciate your work.
Reviewer 2 Report
Comments and Suggestions for Authors
1. In line 13, the words "A not insignificant part" should be replaced by the words "A part that is not insignificant".
2. Line 19: "indicating that the same acidic cluster around QB should response" should replace "should response" with "should respond".
3. Line 217: "The carotenoidless Rba. sphaeroides R-26 was inoculated after incubation" might be clearer if changed to "The carotenoidless Rba. sphaeroides R-26 was inoculated following incubation"。
4. Line 349: "The charge recombination from the P+QA─ state can follow two parallel routes" might be clearer if changed to "Charge recombination from the P+QA─ state can proceed along two parallel routes".
5. Line 473: "The novelty of our investigation was the experimental determination of the drop" might be clearer if changed to "A novel aspect of our investigation was the experimental determination of the drop".
6. In line 493, the words "to establishment" in "The RC shows similar H+ responses (pH-dependence of the free energy states) to establishment" should be replaced by "in response to the establishment".
Comments on the Quality of English Language
the quality of english language is readable.
Author Response
I appreciate the effort and time of the reviewer devoted to evaluate the MS. I'm happy for the positive decision.
I accept and thank for the comments of the reviewer.
All 6 suggestions listed below were taken into account and were introduced into the improved version of the MS.
Reviewer 3 Report
Comments and Suggestions for Authors
This manuscript by Sipka and Maróti is a well written and clear account of the contribution of protonation to relaxation processes that accompany charge separation in the Rhodobacter sphaeroides reaction center. Through the use of reaction center preparations in which the native ubiquinone-10 is replaced by different anthroquinones the authors have investigated the pH dependence and temperature dependence of charge recombination following photoexcitation, and support experimental measurements with computational analysis. The conclusions seem reasonable and are supported by the data. I recommend acceptance for publication subject to some minor revisions.
A general point is that several of the Figures presented in the paper were rather low resolution, such that labelling was difficult to read. Presumably these images can be replaced with higher-resolution versions that are sufficiently sharp that fine detail can be read. This affected, for example, Figures 2, 3 and 4.
Line 61 – change to read “Several lines of evidence have…”
Line 73 – change to read “states relative to that of…”
Line 109 – change to read “structural constraints, as well.”
Line 127 – change to read “The QB domain is rich in protonatable…”
Line 130 – it is not clear what is meant by “supports” in the sentence “and supports the lack of ionizable residues around QA and IA”. This sentence needs rewriting to make the meaning clear.
Line 229 – it is stated that UQ10 was removed from the “native RC from wild-type Rba. sphaeroides” but in the previous section it is stated that the carotenoidless R-26 mutant was used. This contradiction needs clearing up – presumably UQ10 was removed from preparations of R-26 reaction centres?
Line 241 – change to read “The photochemical function of each AQ reconstituted RC…”
Line 523 – change to read “(Asp and Glu) were taken to be 4.5.” (or “assumed to be 4.5.)
Line 616 – change to read “the phytol tail…”
Line 678 – the word “insertion” doesn’t look right here. Do the authors mean “replacement”
Comments on the Quality of English LanguageIn a few places meaning is not quite clear, likely due to word selection. Please see suggested changes to text.
Author Response
I appreciate your time and energy invested to review the MS.
Upon request of the editor, Figs 2-4 can be replaced with higher resolution versions required for the publication.
I accept the majority of your comments listed below and they are now included in the corrected version of the MS. However, I do not see the urgent and unavoidable need of correction of
- line 73 (it is good as it is) and
- line 678 (The expression "insertion" seems to be a proper word as the QA binding pocket is empty (UQ10 is already removed) when AQ is introduced. If UQ10 was there then the word "replacement" would be a better choice - according to my feeling.)